# Quantifying Tree Hydration Using Electromagnetic Sensors

**Lance V. Stott \*, Brent Black and Bruce Bugbee**

Department of Plants Soils and Climate, Utah State University, Logan, UT 84322, USA;
brent.black@usu.edu (B.B.); bruce.bugbee@usu.edu (B.B.)

**\*** Correspondence: lance.stott@usu.edu

**Abstract:** An automated method of determining tree water status would enable tree fruit growers, foresters and arborists to reduce water consumption, reduce orchard maintenance costs and improve fruit quality. Automated measurements could also be used to irrigate based on need rather than on fixed schedules. Numerous automated approaches have been studied; all are difficult to implement. Electromagnetic sensors that measure volumetric water content can be inserted in tree trunks to determine relative changes in tree water status. We performed automated measurements of dielectric permittivity using four commercially available electromagnetic sensors in fruit tree trunks over the 2016 growing season. These sensors accurately measure the ratio of air and water in soils, but tree trunks have minimal air-filled porosity. The sensors do respond, however, to bound and unbound water and the relative change in the output of the sensors thus provides an indication of this ratio. Sapwood is the hydro-dynamically responsive component of trunk anatomy and is nearest the bark. Sensor response improved when the waveguides were exposed to a greater percentage of sapwood. Irrigation-induced increases of approximately 0.5 MPa in stem water potential were associated with 0.5 unit increases in dielectric permittivity. Electromagnetic sensors respond to bound water in trees and thus have the potential to indicate tree water status, especially when the sensor rods are in contact with sapwood. Sensor modifications and/or innovative installation techniques could enable automated measurements of tree water status that could be used to precision irrigate trees.

**Keywords:** water potential; tree water status; TDR; fruit trees; peaches; trunk hydration; electromagnetic sensors; dielectric permittivity; water stress; drought stress

---

## 1. Introduction

Monitoring stem water potential is an effective method for determining tree water status [1], but the method requires much expertise and cannot be automated. Measurements are also limited to the time period nearest to solar noon. An automated method of determining tree water status would allow for the irrigation of orchards based on tree water status, rather than on fixed schedules. Irrigation based on water status could save water, reduce orchard maintenance costs [2–4] and improve water productivity [5,6]. Soil moisture measurements could be effective, but the depth and distribution of tree roots make this method impractical. Several researchers have inserted time domain reflectometry (TDR) and other electromagnetic sensors into wood to determine water content. These sensors have been used to determine the water storage capacity of native conifer trunks [7–12] and to evaluate xylem cavitation [13].

There are several challenges in determining trunk hydration using TDR. Holbrook et al. [14] cautioned that temperature effects in wood could make TDR measurement of trunk hydration more complicated and that waveguide length could also adversely affect these measurements. They suggested using a waveguide length similar to the radius of the stem. A custom calibration equation

relating permittivity to water content may also be necessary [14,15]. In addition, dielectric permittivity is temperature-dependent. When electromagnetic sensors are installed to measure soil water potential, the electronics are generally buried, reducing the potential effects of fluctuations in temperature on both the sensor electronics and the dielectric permittivity. When installed in trees, temperature fluctuations may have a greater effect on water potential readings.

Nadler et al. [16] concluded that TDR could determine stem hydration in lemon and mango [15], but that the signal was too noisy and the system too expensive for managing orchard irrigation. Despite the fact that the system was too expensive for agricultural use [16], its use in research continued. Kumagai et al. [17] found that amplitude domain reflectometry (ADR) sensors bolstered predictions of stomatal conductance. Like TDR sensors, ADR sensors can determine the water content of wood, based on the apparent dielectric permittivity.

The technological advances with TDR (and other electromagnetic volumetric water content) sensors have occurred as Nadler et al. [15] predicted, making the sensors more reliable and cheaper. Garrity [18] suggested using the Decagon GS3 sensor in the trunks of trees to monitor hydration. Using this technique and sensor, Matheny et al. [19] were able to measure the trunk water content of red oak and red maple forest trees. Similar work was done on birch trees by Hao et al. [20], with the exception that the focus was on xylem cavitation. Most recently, Saito et al. [21] demonstrated the utility of electromagnetic sensors in determining the water content of native and invasive trees in arid environments.

In this study, we used four models of electromagnetic soil moisture sensors, inserted into the trunks of fruit trees to test their ability to determine changes in trunk hydration associated with irrigation stress. If electromagnetic sensors can detect changes in trunk hydration, they could be used to automate orchard irrigation and facilitate precision irrigation.

## 2. Materials and Methods

Four models of sensors were used in this study. The Acclima TDR-315 and TDR-315L have nearly identical specifications and are identical in dimensions. These two sensors are grouped together throughout this study as TDR-315(L). The sensors used are shown in Figure 1.

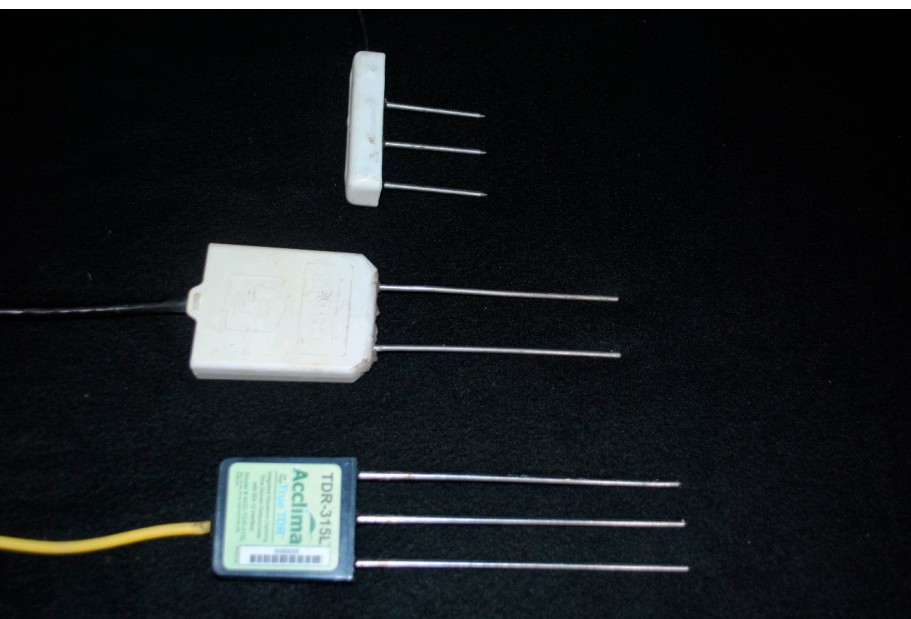

**Figure 1.** Electromagnetic soil moisture sensors used in fruit tree trunks. From top to bottom: Decagon Devices GS3, Campbell Scientific CS655, Acclima TDR-315(L). TDR-315 and 315L are nearly identical in specification and were treated as the same sensor for the study. Only the TDR-315L is pictured here.



The TDR-315(L) and the Decagon GS3 have three waveguides, while the Campbell Scientific CS655 has two. (Table 1). The manufacturer-listed volume of influence ranges from 100 mL to nearly 1.5 L, but volume of influence varies with water content, target medium and installation methods and sensors should be calibrated accordingly [22]. Frequencies also differ greatly between sensors, ranging from 70 MHz to 4 GHz. Probe length for GS3 sensors is 5 cm while the CS655 and TDR-315(L) are approximately twice as long, 12 and 15 cm, respectively. The effective frequency of each sensor will vary with the characteristics and water content of the measured medium [23].

**Table 1.** Description of electromagnetic soil moisture sensors. Sensors varied in waveguide length, number of waveguides, volume of influence and frequency. The TDR-315 and TDR-315L are nearly identical and were grouped for this study.

|  | Decagon Devices GS3 | Campbell Scientific CS655 | Acclima TDR-315 | Acclima TDR-315L |
|---|---|---|---|---|
| No. of waveguides | 3 | 2 | 3 | 3 |
| Waveguide length | 5 cm | 12 cm | 15 cm | 15 cm |
| Volume of influence | 160 mL | 3600 mL | 100 mL | 100 mL |
| Frequency | 70 MHz | 78 MHz | 3.5 GHz | 4 GHz |
| Output | Permittivity and VWC | Permittivity and VWC | Permittivity and VWC | Permittivity and VWC |

Prior to installation, the temperature sensitivity of each type of the sensor was tested in the lab. Replicate sensors were suspended in the air in a dark growth chamber for ten days while the temperature changed gradually between 10 °C and 35 °C over the first 12 h and from 35 °C back to 10 °C over the second 12 h of each day.

Sensors were installed in a 12-year-old 'Suncrest' Peach orchard on a seedling rootstock. The trees were trained in an open vase system with bare ground under the trees and grass between rows. Trees were spaced 3.7 m apart within row, with rows on 6.1 m centers. Pilot holes just larger than the probes of each respective sensor were drilled with a jig to ensure proper alignment. Sensors were then installed using a rubber mallet, if necessary. In preliminary studies, sensor waveguides were installed through the center of tree trunks to maximize waveguide contact with wood. However, cross sections of fruit trees revealed that many trees had thin layers of sapwood surrounding the heartwood. Sapwood is where most of the water transfer in a tree trunk occurs, therefore, sensors were installed at an angle in order to maximize waveguide contact with the sapwood—despite portions of the waveguide protruding completely through the trunk in some cases.

Seven TDR-315(L), seven CS655 and four GS3 sensors were installed on 13 June 2016 into nine different trees. Trees were next to each other in two adjacent orchard rows to facilitate datalogger installation and wiring. Each of the nine trees had two sensors. Four additional GS3 sensors were installed in two additional trees after the initial installation. Various combinations of two sensors were used for most trees, but two trees had two GS3 sensors. When possible, sensors were installed on the north side of the peach trees to avoid potential heating from solar radiation. Some sensors were installed in the west side because of trunk geometry (Figure 2A). All trees received the same irrigation treatment.

Parts of the waveguide near the sensor head were not in the tree for some sensors (Figure 2A). In addition, the CS655 and TDR-315(L) probes were long enough to go completely through the tree trunk in some cases (Figure 2B). This exposed portion of the waveguide would reduce the signal from the sensor because the part of the waveguides exposed to the air would sense the permittivity of air which is 1. This exposure to air would attenuate the signal in proportion to the fraction of the rods in air. Because we are interested in relative changes, and because the fraction of the rods in the air is constant, this was not a significant concern.

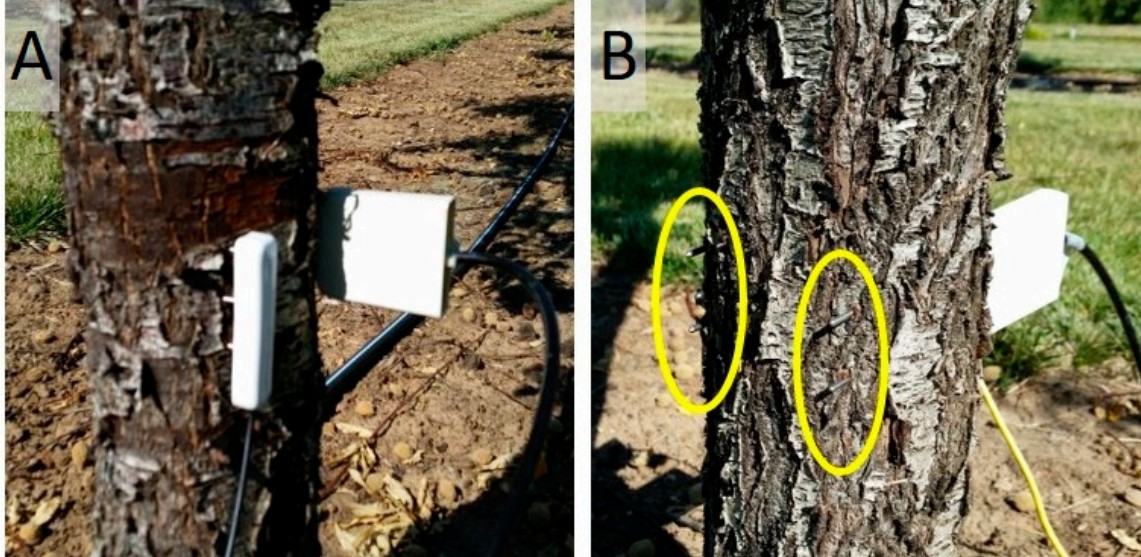

**Figure 2.** (**A**) GS3 and CS655 installed in a peach tree trunk revealing some exposed waveguide near the sensor head. (**B**) CS655 and TDR-315(L) sensors protruding through a peach tree trunk.

To induce drought-stress, the entire orchard was not irrigated for several weeks and then the soil moisture was completely replenished. We used a pressure chamber to provide a reference for tree hydration. Water was withheld from all trees until stem water potential readings approached −2.0 MPa. Then trees were irrigated until the soil moisture was completely replenished (stem water potentials near −1.0 MPa). Four dry-down and irrigation cycles occurred during the growing season. When irrigated, water was applied to replace water loss by evapotranspiration (ET), as calculated by the Penman-Monteith equation from data obtained from a weather station in the orchard. Irrigation varied between 30 and 50 mm per week, depending on the time of the year.

Stem water potential using a pressure chamber [1,24] was evaluated three times per week to develop a correlation between stem water potential and trunk permittivity. Four leaves from each tree were enclosed in Mylar bags in the morning and allowed to equilibrate with the main stem before measurements were made around solar noon. Three measurements were made on each tree on each sampling day. The three measurements were averaged to determine the SWP of each tree for that sampling day. Wherever possible, the SWP measurement from the two days immediately preceding the irrigation event were averaged to determine a pre-irrigation SWP for each tree. Likewise, the SWP measurement of each tree was averaged for the two sampling days immediately following the irrigation event to determine a post-irrigation SWP.

At the end of the season, the peach trees were cut down. A 40-cm long section of the trunk in which the sensors were installed was excised and brought to the lab for further analysis. Each cut end of each peach trunk section was covered with petroleum jelly to prevent evaporation. All trunk sections were then placed in a dark growth chamber to test for temperature sensitivity. Two thermocouples were installed in each trunk section. Holes were drilled in the trunk near each sensor to a depth similar to that of the waveguides. The growth chamber ramped steadily from 10 °C to 35 °C over 12 h and then ramped back down to 10 °C over the next 12 h. Data from all electromagnetic sensors were collected with a datalogger (Campbell Scientific, model CR1000).

After the temperature sensitivity test, the top end of each trunk section was re-cut and photographed to illustrate the proportions of sapwood and heartwood. A visual assessment of the proportion of each sensor that was in heartwood, sapwood or outside the bark was performed using a ruler to superimpose a line on the top of each trunk section representing the path of the waveguides. The length of the waveguide in each part of the trunk was measured. Pictures of the trunk sections are shown below with black dotted lines indicating the waveguide locations (Figure 3).

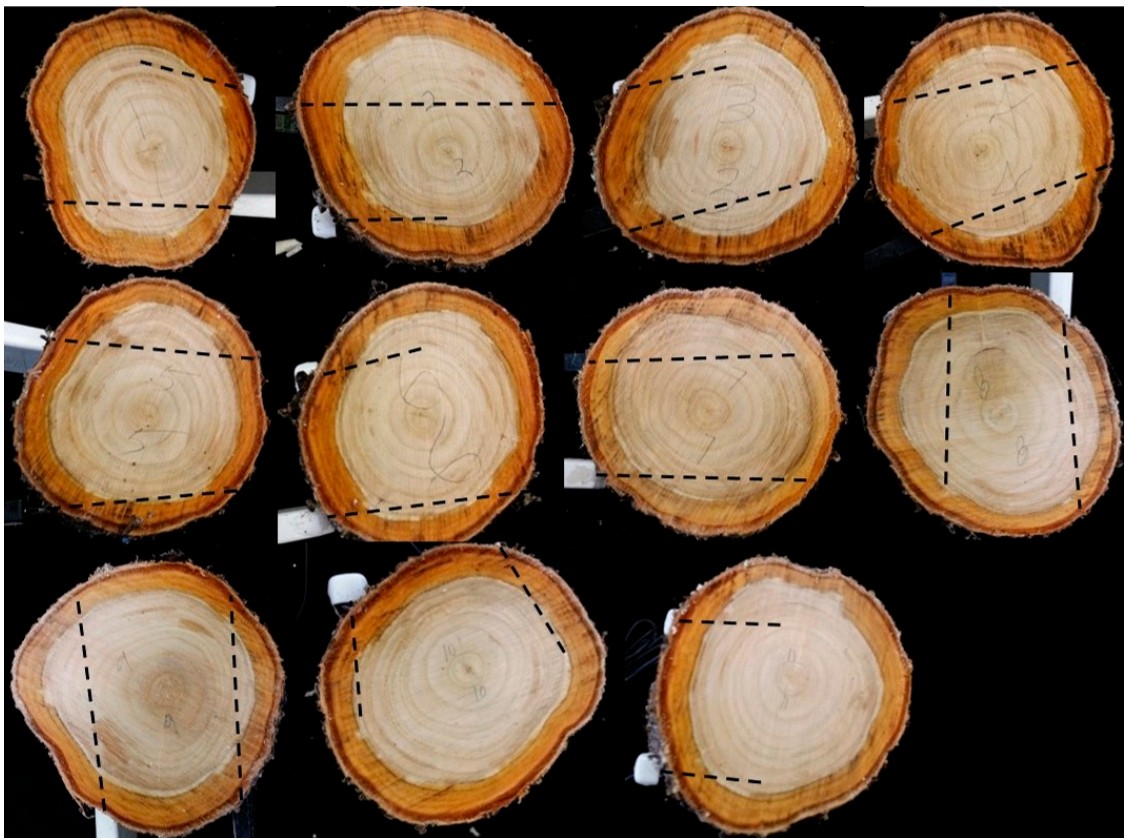

**Figure 3.** Sapwood and heartwood ratios for all excised peach trunk sections. Sapwood was comprised of the outer four or five annular rings and was about 3 cm thick (darker colored wood near bark). Black dotted lines indicate waveguide locations.

## 3. Results

GS3 permittivity output generally decreased between irrigations and recovered following them. A degree of recovery was immediately detectable, but recovery continued for four or five days following irrigation. Then permittivity values began to decline again (Figure 4A).

CS655 permittivity values decreased between every irrigation and recovered following the irrigation with the exception of a single sensor which did not respond as expected between the irrigations on July 5 and July 28. Similar to the GS3, permittivity values showed immediate recovery, but increased over the following four to five days before declining again (Figure 4B).

TDR-315(L) permittivity responded similarly to the other two sensors. After the 5 July irrigation, all sensors detected trunk dehydration between irrigations and recovery of trunk hydration immediately after irrigation with continued recovery for four to five days afterward (Figure 4C).

Changes in permittivity before and after irrigations were small (<1 permittivity unit) for each model of sensor. CS655 sensors recorded the largest difference in permittivity, followed by GS3 sensors and TDR-315 sensors (Table 2).

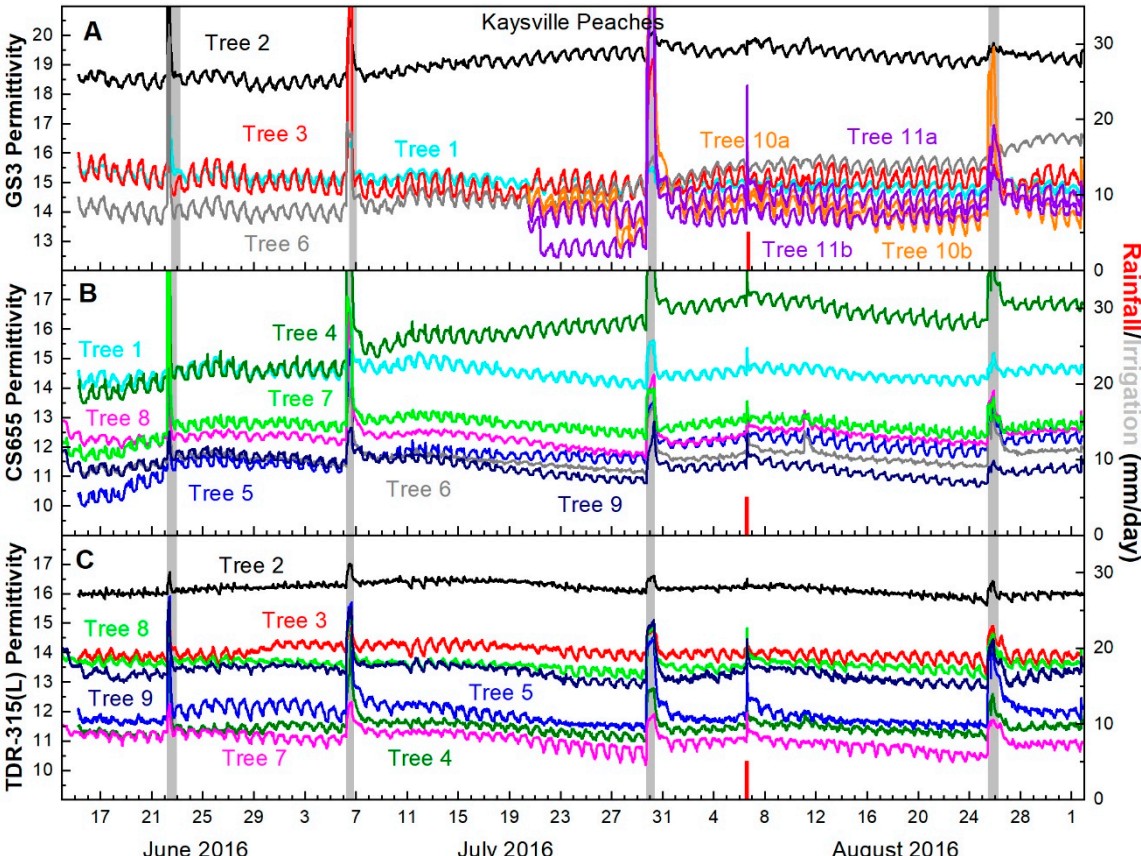

**Figure 4.** Peach trunk permittivity from (**A**) GS3 sensors (**B**) CS655 sensors and (**C**) TDR-315(L) sensors. Line colors indicate trees, not sensor models e.g., Tree 2 had one GS3 (black line in Figure 4A) and one TDR-315 (black line in Figure 4C). The single rainfall event on 5 August is indicated with a red bar, while irrigations are indicated with gray bars. Irrigation bars indicate date of irrigations and volumes varied by need.

**Table 2.** Stem water potential and permittivity before and after differences for four irrigations. Stem water potential increases of 0.5 MPa corresponded to permittivity increases ranging from 0.27 to 0.39 units. *p*-Values are for a paired *t*-Test.

| | **Stem Water Potential Increase** | **Permittivity Increase** | | |
|---|---|---|---|---|
| | (MPa) | TDR-315(L) | CS655 | GS3 |
| Mean | 0.50 | 0.25 | 0.39 | 0.27 |
| SD | 0.21 | 0.19 | 0.20 | 0.21 |
| *p*-Value | *p* < 0.01 | *p* < 0.01 | *p* < 0.01 | *p* < 0.01 |
| Reps | 11 | 7 | 7 | 8 |

Stem water potential varied and $r^2$ values for correlations with trunk permittivity ranged from $r^2 = 0$ to $r^2 = 0.17$ for GS3 sensors, from $r^2 = 0.01$ to $r^2 = 0.26$ for CS655 sensors and from $r^2 = 0.03$ to $r^2 = 0.29$ for TDR-315(L) sensors.

Despite the fact that the relationship between contact with the sapwood and sensor response was not overly robust ($r^2 = 0.41$, $p = 0.13$ for TDR-315(L); $r^2 = 0.27$, $p = 0.23$ for CS655; $r^2 = 0.20$, $p = 0.27$ for GS3) for any of the sensors, the sensor response increased as the percentage of the sensor waveguide in contact with the sapwood increased (Figure 5).

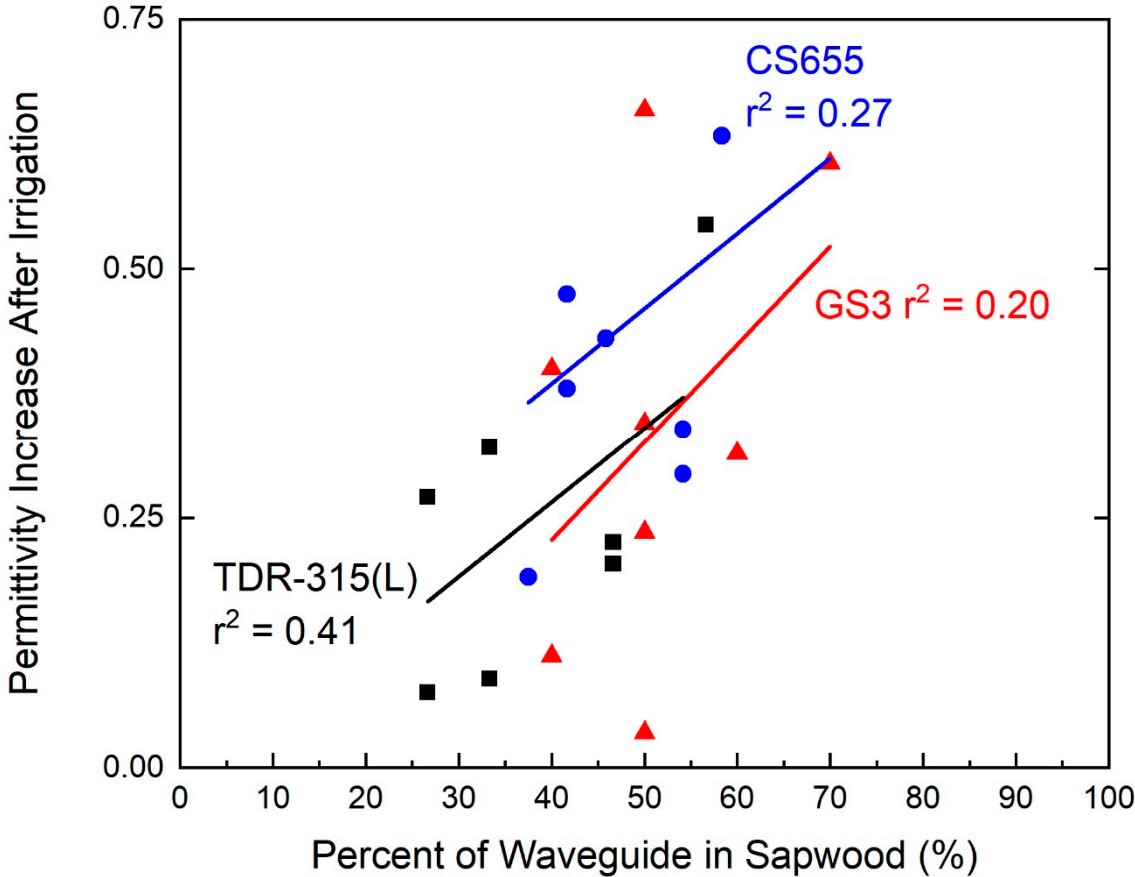

**Figure 5.** Relationship between waveguide contact with sapwood and changes in permittivity before and after irrigations. Sapwood contact was directly correlated with permittivity increase.

Diurnal fluctuations in permittivity can be observed in the sensor data. Pre-installation tests of each sensor's electronics revealed very little sensitivity to temperature for all sensors. Of the ten sensors tested (2 GS3s, 3 CS655s, 2 TDR-315s and 3 TDR-315Ls), only four sensors responded to temperature. For the two GS3 sensors, the response was +0.0011 and +0.0051 permittivity units per degree Celsius. For the two TDR-315 sensors the response was even smaller (−0.00014 and +0.00015 permittivity units per degree Celsius). All three CS655s and all three TDR-315Ls tested show no response to temperature (see supporting information). Temperature sensitivity tests of the sensors installed in the cut sections of the trees revealed very little temperature sensitivity. The average slope for GS3 sensors was −0.0165 units of permittivity per degree Celsius while the slopes for the CS655 and TDR-315(L) were −0.0012 and 0.015, respectively.

For field data, the average slope of temperature and permittivity relations for all GS3 data is −0.037 units of permittivity per degree Celsius. The slopes for the CS655 and the TDR-315(L) are −0.012 and −0.003, respectively. This is much less than the expected value.

Focusing on one week's data revealed that permittivity decreased in the afternoons and increased during the night and morning. Fluctuations of nearly one permittivity unit were observed for some sensors (Figure 6).

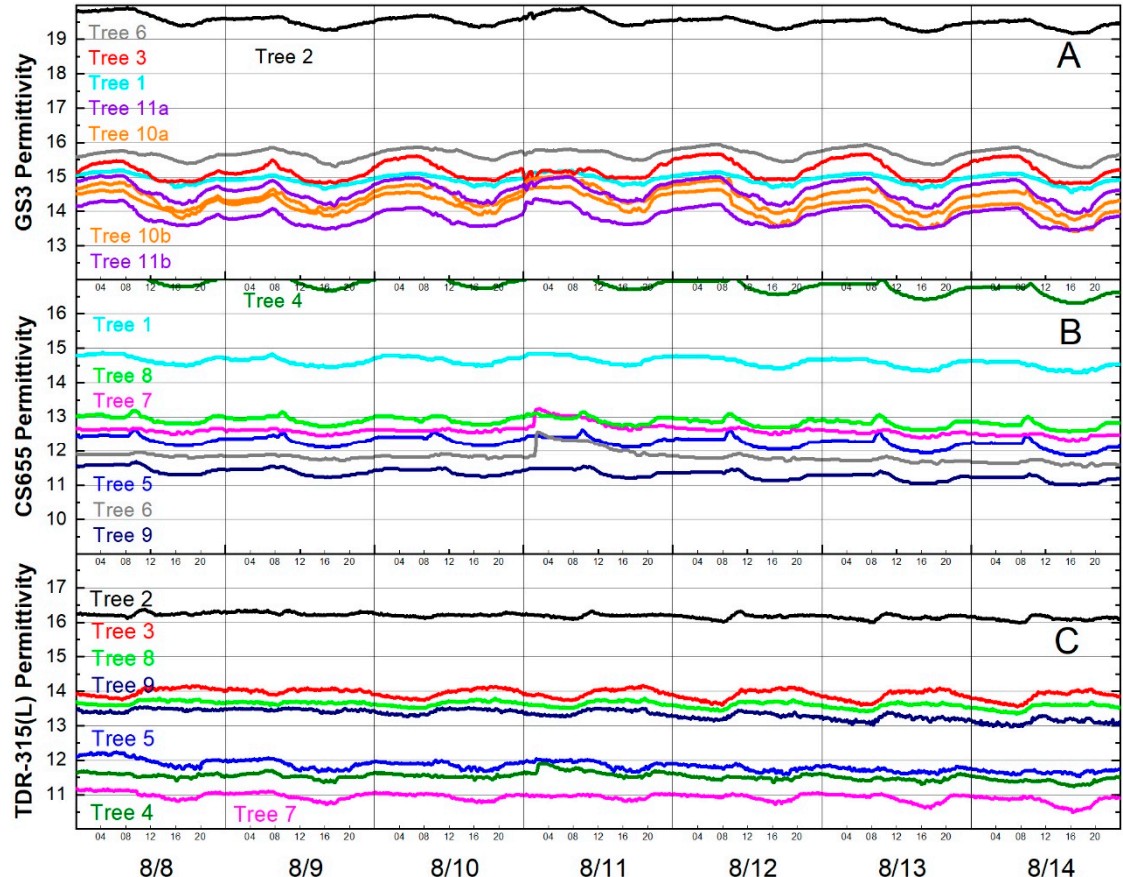

**Figure 6.** Diurnal fluctuations in permittivity for the week of 8 August 2016 for (**A**) GS3 sensors, (**B**) CS655 sensors and (**C**) TDR-315(L) sensors. Line colors indicate trees, not sensor models, e.g., Tree 2 had one GS3 (black line in (**A**)) and one TDR-315 (black line in (**C**)).

## 4. Discussion

Ultimately, a strong relationship between trunk permittivity and SWP would indicate that trunk permittivity obtained with soil moisture sensors is a suitable replacement for stem water potential measurements. The actual permittivity of a tree's trunk is less important than being able to detect a change in permittivity measurements before and after an irrigation (or precipitation) event. Trunk permittivity values were small and variable for all types of sensors used, but correlations between trunk permittivity and SWP were significant. There were ideal weather conditions in 2016 for testing for these differences since there was only one small rainfall event (5 mm between 13:00 and 14:00) on 6 August. The lack of precipitation allowed for larger differences in tree water status. Still, since these sensors are designed to be installed in soils, it is not surprising that there are challenges in using them to sense the hydration of tree trunks.

In preliminary studies (2014 and 2015), sensors were placed straight into the trunk so as to ensure the entire length of the sensor waveguides was in the tree trunk, but contact with sapwood improved the sensor response. In this study, sensors were placed at an angle to maximize contact with sapwood, but, at the end of the study, these peach trees had a large proportion of heartwood. Trees were approximately 15 cm in diameter with about 3 cm of sapwood surrounding the heartwood. The sapwood was only approximately 30% to 35% of the 180 cm$^2$ trunk cross-sectional area. In addition, fruit trees tend to have smaller trunk diameters and, in some trees, the waveguides of CS655 and TDR-315(L) sensors protruded through the trunks. Because of this, the part of the sensor in the air would detect a permittivity of 1, diluting the signal. This variation in sapwood contact could explain why the signal detected by the sensors was smaller than expected. Nevertheless, placing the sensors

in a manner to maximize contact with sapwood is beneficial since the stem water potential increase measured by the sensors was directly correlated with the percent of the waveguide that had contact with sapwood.

Saito et al. [21], Matheny et al. [19] and Hao et al. [20] all found that electromagnetic sensors could reliably measure tree trunk hydration. Perhaps the percent sapwood contact could be the reason why we were not able to entirely corroborate their results. Doing an evaluation of sapwood thickness using sample cores of the target species as Bovard et al. [25] and Matheny et al. [19] did could help to maximize sapwood contact. It appears that some customization of sensors might also be required to allow them to be installed properly in fruit trees. Contact with sapwood may be more important than the actual length of the waveguides since all of the sensors responded to increased sapwood contact even though the GS3 waveguides are about half as long. Maximizing exposure to sapwood seems to be essential and may be accomplished through sensor modification or selection and installation methods. Still, electromagnetic sensors are capable of detecting changes in tree trunk permittivity, though small.

The small sensor signal demonstrates the difficulty of making these measurements. However, the benefits of an automated method of measuring tree water status cannot be understated. These measurements might be improved by optimizing installation techniques so as to maximize contact with the sapwood. Sapwood contact might be maximized by installing the sensor waveguides at a downward shallow vertical angle. Modifying sensor waveguide design could also improve the hydration signal from trees, e.g., developing sensors where the waveguides could be mounted vertically in grooves on the exterior of, or wrapped around, the trunk of the tree. This could be done with little disruption of the vascular conductivity.

Despite some evidence of temperature influence on sensor output, it appears that the sensors are also capable of detecting small diurnal fluctuations in trunk water status. These fluctuations could have been caused by changes in tree hydration, but could be the result of temperature effects on the permittivity of water. All sensors reported a similar season-long minimum temperature, but the season-long maximum temperature recorded by the TDR-315(L) was approximately 10 degrees higher than that recorded by GS3 or CS655 sensors. The sensor body of both the GS3 and the CS655 is white, while the sensor body of the TDR-315(L) is black. This difference in color likely explains why the maximum temperatures vary, while the minimum temperatures do not.

The permittivity of water changes with temperature. The relationship of water and permittivity can be found using:

$$\varepsilon = 87.740 - 0.4008t + 9.398 * 10^{-4}t^2 - 1.410 * 10^{-6}t^3,$$

where $\varepsilon$ permittivity and $t$ is temperature in degrees Celsius [26]. The approximate range of temperatures in this study is 5 °C to 55 °C. This part of the curve relating permittivity and water can be approximated with a linear equation with a slope of −0.36 and an $r^2 = 0.99$. In other words, the permittivity of water decreases by 0.36 for every 1 °C increase in temperature from 5 °C to 55 °C.

For all sensors, the average slope of temperature and permittivity relationships is much less than the expected value. One possible explanation is that the temperature inside the tree trunk is more stable than the temperature detected in the sensor head. However, a lab temperature sensitivity test indicated that wood temperature at the depth of the sensor lagged air temperature by only one or two degrees (data not shown).

Still, other factors in the field such as solar radiation, sap flow rates and wind could affect the temperature of both the sensor head and the wood. This could result in a greater difference in temperature between the two readings. The only way to characterize this difference in the field would be to install a thermocouple or thermistor in the tree near the sensor waveguides to simultaneously monitor differences between wood temperature and sensor body temperature. Even so, temperature effects on permittivity were less than what would be expected if the sensors were only "seeing" bulk water which indicates that bound and unbound water play a role in the response of permittivity to temperature in wood as has been suggested in soils [27,28].

Assuming sensor electronics are minimally sensitive to temperature, we would expect that, for each 1 °C increase in temperature, permittivity values would drop by 0.36 units. However, temperature changes also affect the electrical conductivity of water (EC), which, in the case of the CS655 sensors, affects the period value, and, consequently permittivity (Ritter, personal communication). EC increases by 2% for each degree Celsius increase in temperature in the case of the CS655. These two interacting factors make it difficult to correct sensor output for the effect of temperature on permittivity. Further, the interacting effects of temperature on water bound to solid surfaces and on bulk soil water create a complex interaction where an empirical temperature correction is impossible [28]. Or and Wraith [27] suggested that the thickness of the layer of water bound to solid surfaces is affected by temperature and offered corrections based on soil specific surface area and water content. These parameters can be estimated from soil texture, but in order to employ similar corrections in tree trunks, the wood specific surface area and water content of each tree species would need to be estimated.

Since the effect of temperature on EC can also affect CS655 permittivity, this effect must also be explored in order to provide evidence that the sensors were able to detect a real diurnal fluctuation in permittivity. The average slope of temperature and EC relations for the CS655 was less than 0.02% per degree Celsius—much less than the expected 2% change. Thus, because the actual slopes of the relationship between temperature and EC are much less than the expected slope, there is no apparent need for temperature correction based on its effect on EC. This may be partly explained by the fact that the EC values detected by the CS655 (approximately 0.05 dS/m) are very low (EC of tap water in the area is approximately 0.34 dS/m). Because the EC measurements are low, they likely induce a minimal effect on permittivity as temperature increases.

The fact that permittivity readings seem to be temperature-stable for each type of sensor added to the fact that permittivity decreases during the day and increases during the night provides evidence that the sensors are capable of detecting diurnal fluctuations in tree trunk hydration.

Still, our work confirms Holbrook's (1992) caution about temperature sensitivity. The difference in temperature between the sensor body and the wood could affect measurements from the electromagnetic sensors, but this is likely not as great as the effect of temperature-sensitive electronics. At the very least, a sensor whose electronics are stable is a requirement for this type of measurement. Diurnal changes in trunk hydration, while interesting, may be of less value than the daily mean values in terms of scheduling irrigation based on tree water status—particularly if there is uncertainty about temperature effects on measurements.

The large range of temperatures detected by the sensors suggest that insulating the sensors as Saito et al. (2016) did might be of benefit. Despite the insulation, daily temperatures in their study fluctuated approximately 10 °C. The daily fluctuations in our study were approximately 20 °C for GS3 and CS655 and approximately 30 °C for TDR-315(L). Even with this large diurnal temperature change, the effect of temperature on permittivity readings was small, suggesting that insulation may not be necessary.

Even though temperature sensitivity was small, we would have expected a greater response from the sensors. The small response could be explained by lack of sapwood contact combined with exposure of portions of some sensor waveguides to air. In order for electromagnetic sensors to be used for irrigation automation, the signal to noise ratio would need to be reduced. All sensors had similar challenges and produced similar output. However, the cost of automated measurements with electromagnetic sensors is much less than the cost of manual measurement of stem water potential. Scholander-type pressure chambers currently sell for approximately $3200. GS3 and CS655 sensors cost about $250 each; TDR-315 sensors cost about $400; and TDR-315L sensors cost about $300 each. All sensors require a datalogger, which sells for approximately $1000 when equipped with cell phone communication capabilities. One datalogger can monitor at least 40 sensors (with SDI-12 output and up to 1000 feet of sensor cable per channel). The initial cost of an automated electromagnetic sensor system with 8 sensors is similar to the initial purchase of a pressure chamber.

However, labor to make manual pressure chamber measurements of stem water potential is about $10 per measurement (assuming $40 per hour and four measurements per hour). To measure eight trees per day is a labor cost of approximately $80 per day. The labor cost of the automated system would be minimal after the initial installation. Significant labor cost savings and the benefits of automated measurements warrant further testing and development on electromagnetic measurements of tree hydration.

Improved installation techniques may help to improve the signal. In addition, our data suggest that the small signal detected by these sensors could be improved by optimizing waveguide shape and/or length to maximize exposure to sapwood.

**Supplementary Materials:** The following are available online at http://www.mdpi.com/2311-7524/6/1/2/s1.

**Author Contributions:** Conceptualization, B.B. (Brent Black) and B.B. (Bruce Bugbee); methodology, L.V.S., B.B. (Brent Black) and B.B. (Bruce Bugbee); software, L.V.S.; validation, L.V.S., B.B. (Brent Black) and B.B. (Bruce Bugbee); formal analysis, L.V.S.; investigation, L.V.S.; resources, B.B. (Brent Black) and B.B. (Bruce Bugbee); data curation, L.V.S.; writing—original draft preparation, L.V.S.; writing—review and editing, B.B. (Brent Black) and B.B. (Bruce Bugbee); visualization, L.V.S., B.B. (Brent Black) and B.B. (Bruce Bugbee); supervision, B.B. (Brent Black) and B.B. (Bruce Bugbee); project administration, B.B. (Brent Black) and B.B. (Bruce Bugbee); funding acquisition, B.B. (Brent Black) and B.B. (Bruce Bugbee). All authors have read and agreed to the published version of the manuscript.

**Funding:** This research was funded by the Utah Department of Agriculture and Food, Specialty Crop Block Grant and by the Utah Agriculture Experiment Station (journal paper number 9258).

**Conflicts of Interest:** The authors declare no conflict of interest.

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
