# Peer review of "Quantifying Tree Hydration Using Electromagnetic Sensors"

_horticulturae, doi:10.3390/horticulturae6010002_

Round 1

Reviewer 1 Report

Dear authors, I finish the review of you manusccript and find it interesting, well written and designed. In my opinion, you made really a nice work addressing a topic of interest for researchers involved in determining plant water status. I recommend acceptance with very few changes. I have, however, some questions I would like authors to address. My questions follow their sentences in cursive text

Rather than trying to ensure that the entire length of the sensor probe was inside the tree, installation focused on trying to get as much of the probe in the sapwood of the tree as possible.

Can you explain a little for readers not familiar with tree trunk anatomy?

Seven TDR-315(L), seven CS655 and four GS3 sensors were installed on June 13th

June 13th 2019? One sensor per tree? Different trees for different sensors? Same trees holding all sensors? How were the trees selected? Were they under different irrigation treatments?

Leaves were enclosed in a Mylar bag..

For how long?

Wherever possible, the two readings…

What two readings? Performed in different days or in different trees? Please, explain (also for L 212) and give reasons for.

Then permittivity values began to decline again

Not clear to me. For instance, tree 2 in July, and 6 in June. As stated in Line 134 changes in permittivity seem very small.

The relationships between SWP and permittivity (with all sensors) estimated by regression analyses seem weak.

Finally, can you in your discussion address the matter and speculate about representativeness and how this depends on the number of sensors and then about the cost comparing with SWP measurements?

Author Response

Thank you for your thorough review of the submission.  We have addressed the concerns you presented and it is uploaded as a word document.  The highlighted text is our response to each issue you pointed out.

Reviewer 2 Report

Review

This Article offers an empirical evaluation of Tree Hydration sensors, namely using Electromagnetic soil moisture sensors. This study may prove relevant for farmer and the generic public. However, it raises even more questions than answers, which makes its interpretation and use quite limited.

This Article presents a quality English writing.

Abstract

The abstract is consistent with the content of the article.

Introduction + Urbanization and Planning of Green Spaces in Beijing

The introduction is clear. I would suggest that it should include the potential effect of temperature to avoid a later introduction of the topic in the discussion chapter (ex. Lines 184-190).

Methodology

The methodology is well described and the images provide further information regarding the details of the experiments layouts.

Results + Discussion

Results are rather confusing as there is no clear differentiation between the results in this section and the later addition of data in the discussion chapter. I would recommend that the results should be presented at once (first), to then be discussed (after) in the following section.

The authors concentrate mainly on explaining the potential methodological aspects that may have influenced the results and their potential bias.

I would suggest that additional analysis could include the statistical study of the similarities in the behavior of the different sensors tested (ex. Correlation or other data differences analysis) in order to assess if the sensors have a similar performance.

Detailed corrections

Line 48 Vs Line 53 – Please avoid ambiguity regarding the number of sensors evaluated in this study. Clarify or use the same number.

Line 60 – Use “three” instead of “3”

Line 65 – “while the CS655 and TDR-315(L) are approximately twice as long” please provide a more accurate information as it clearly doesn’t expressed the differences in the length of the sensors.

Overall, I think that this Article may provide useful information for future research and may suggest limitations in the use of Electromagnetic soil moisture sensors; however, its results may fail to provide the needed consistency to ultimately determine the use of this kind of sensors.

For the overstated reasons I am recommending that your article is considered for major revision, in the hope that the Authors may clarify some of the existent doubts further ahead in this revision process.

Kind regards,

Author Response

Thank you for your thorough review of our submission.  We have uploaded point by point responses in a word document.  Highlighted text is our response.  

Reviewer 3 Report

This is an interesting study testing electromagnetic sensors for soil volumetric water content on trees’ stem water status. The results indicated potentials of the sensors in capturing changes in stem water with apparent limits due to contact area and temperature sensitivities. The data generally support the conclusion, but I have some comments and suggestions on the manuscript.

            My principal concern is with discussion section. Several paragraphs in the discussion section introduced new results, and some of them without showing the data. For example, a paragraph about the permittivity of water changes with temperature (L184-190) need to be mentioned in method section and results section possibly along with a figure. The next two paragraph (L191-209) also contain some information which might fit better in results section. Especially, slopes of temperature and permittivity should be introduced in the results section before further discussion on the matter. Lab test results for the relationship between temperature and permittivity also need to be provided as results or at least as supporting information.

            In Figure 4, including all trees with different colors may not be efficient for readers to interpret the main idea of the figure. I recommend to use mean values of permittivity for each sensor type with some uncertainty ranges. Current figure can be provided as supporting information.   

            Similar to Figure 4, Figure 6 is hard to interpret. In addition, it needs to be a result, not a discussion material. Readers generally don’t expect new results in the discussion section. It is hard to see the diurnal cycles. I suggest diurnal cycles from average through time) or average among trees with the same sensor type.   

Minor comments:

L167: What are the results from the three references? Readers may not remember them.

Author Response

Thank you for your thorough review of our submission.  Point by point responses are provided here in a word document where our explanations are highlighted.

Reviewer 4 Report

The manuscript ''Quantifying Tree Hydration using Electromagnetic Sensors''

is a contribution in dealing  with the issue of irrigation automation that assume great importance nowadays. The manuscript is well written, I have only few suggestions. In particular:

L11: Replace ''We made automated'' with ''We performed automated''

L60: Replace ''wave guides'' with ''waveguides''

L70: How long lasted the experiment and in which year was performed?

L121/125/128: add the colour line to ''(figure 4)''

L208/210: ''However'' is a repetition. Start the sentence at L210 with a different word

L239: The part ''These fluctuations....of water.'' does not belong to figure caption but discussion.

L257: I suggest to highlight the importance of this study and its concrete application in the field of irrigation, in the manuscript this does not stand out. 

Author Response

Thank you for your thorough review of our submission.  We have provided a point by point response in the attached word document.  Our responses are highlighted.

Round 2

Reviewer 2 Report

Dear Authors,

Most of the previous concerns have been answer, although this article still raises many questions regarding some methodological aspects, most of them acknowledged by the authors, which may be seen as a strong limitation to the use of the studied sensors. However, as it may prove interesting for further developments in this field, I am suggesting this article to be accepted

Kind regards,

Author Response

Thank you for your second review.  As you said, we know that these sensors are not ready for field deployment, but we, too, feel that there is potential for further development here.

Reviewer 3 Report

The manuscript seemed to have more solid story after the revision. I still have some minor comments and suggestions on the manuscript.

In Discussion section, the sensitivity of permittivity to temperature may need to be described in Methods section and Results section. Was the result of the lab test in lines 257-259 provided as a supporting information? I cannot find a supporting information, and it is still stated as “data not shown” in the main text.

Minor comments:

L183-184: Why the data are “not shown”?

L213: Be specific. What is the average proportion of heartwood (in area- or volume-based)?

Author Response

We have added information about the sensitivity of permittivity to temperature in the Methods and Results section as suggested.  We have also provided the resulting figures in the supporting data as well as giving numerical results in the text.  The temperature response regressions are all less than 0.01 units of permittivity per degree Celsius, which is why we did not show the data initially.

We also added information about the average proportion of heartwood in terms of percent trunk cross sectional area (TCSA).

Thank you for your thorough review of our manuscript.